health and disease and epidemiology, genetics, ecology

toxicology, malaria, synergism, CRISPR/Cas9, insecticide resistance management, *Drosophila*

**Authors for correspondence:**
Vassilis Douris
e-mail: vdouris@imbb.forth.gr
John Vontas
e-mail: vontas@imbb.forth.gr

[†]These authors contributed equally to this work.
[‡]Present address: School of Pharmacy, University of East Anglia, Norwich Research Park, Norwich NR4 7TJ, UK.

# 'What I cannot create, I do not understand': functionally validated synergism of metabolic and target site insecticide resistance

George-Rafael Samantsidis[1,2,†], Rafaela Panteleri[1,2,†], Shane Denecke[1], Stella Kounadi[1,2,‡], Iason Christou[1,2], Ralf Nauen[3], Vassilis Douris[1,4] and John Vontas[1,5]

[1]Institute of Molecular Biology and Biotechnology, Foundation for Research and Technology Hellas, 100 N. Plastira Street, 70013 Heraklion, Crete, Greece
[2]Department of Biology, University of Crete, Vassilika Vouton, 71409 Heraklion, Crete, Greece
[3]Bayer AG, CropScience Division, R&D Pest Control, 40789 Monheim, Germany
[4]Department of Biological Applications and Technology, University of Ioannina, 45110 Ioannina, Greece
[5]Laboratory of Pesticide Science, Department of Crop Science, Agricultural University of Athens, 118 55 Athens, Greece

G-RS, 0000-0002-8279-2114; SD, 0000-0002-7291-1394; VD, 0000-0003-4608-7482; JV, 0000-0002-8704-2574

The putative synergistic action of target-site mutations and enhanced detoxification in pyrethroid resistance in insects has been hypothesized as a major evolutionary mechanism responsible for dramatic consequences in malaria incidence and crop production. Combining genetic transformation and CRISPR/Cas9 genome modification, we generated transgenic *Drosophila* lines expressing pyrethroid metabolizing P450 enzymes in a genetic background along with engineered mutations in the voltage-gated sodium channel (*para*) known to confer target-site resistance. Genotypes expressing the yellow fever mosquito *Aedes aegypti Cyp9J28* while also bearing the $para^{V1016G}$ mutation displayed substantially greater resistance ratio (RR) against deltamethrin than the product of each individual mechanism ($RR_{combined}$: 19.85 > $RR_{Cyp9J28}$: 1.77 × $RR_{V1016G}$: 3.00). Genotypes expressing *Brassicogethes aeneus* pollen beetle *Cyp6BQ23* and also bearing the $para^{L1014F}$ (*kdr*) mutation, displayed an almost multiplicative RR ($RR_{combined}$: 75.19 ≥ $RR_{Cyp6BQ23}$: 5.74 × $RR_{L1014F}$: 12.74). Reduced pyrethroid affinity at the target site, delaying saturation while simultaneously extending the duration of P450-driven detoxification, is proposed as a possible underlying mechanism. Combinations of target site and P450 resistance loci might be unfavourable in field populations in the absence of insecticide selection, as they exert some fitness disadvantage in development time and fecundity. These are major considerations from the insecticide resistance management viewpoint in both public health and agriculture.

## 1. Introduction

The prevention of vector-borne diseases and the protection of agricultural production largely relies on the control of pest insects through the use of insecticides. However, insects display a striking ability to develop resistance, an intriguing evolutionary adaptation to a very fast environmental change, with dramatic consequences. For example, the number of malaria cases increased in 2018 after many years of decline, indicative of a failure of pyrethroid based intervention strategies [1].

To mitigate against the failure of insecticide-based control tools, the mechanisms by which insects have evolved resistance must be elucidated and understood. Mutations at the insecticide target site which reduce insecticide binding affinity,

and metabolic detoxification which inactivates and sequesters insecticidal active ingredients, are the most common mechanisms of insecticide resistance [2]. However, it has been widely hypothesized that it is only the synergism of different mechanisms in the same insect population that causes a real operational control failure in many cases [3–6]. This has important ramifications on insecticide resistance management (IRM) strategies. For example, the synergist piperonyl butoxide when incorporated into bednets seems to restore their efficacy even in areas with fixed target site resistance alleles [7], while the value of molecular diagnostics for IRM might be different depending on the presence or absence of additional mechanisms in the same mosquito population [8,9]. Although a putative synergistic epistasis of the metabolic and target site resistance loci has been considered [6], it has only been tested by crossing lines with different resistance factors together in order to see their effect. However, this process introduces a large amount of unrelated genetic variation, which complicates inferences drawn about the specific loci being studied. While there have been several efforts to isolate the contribution of resistance alleles by introducing them into model organisms like *Drosophila*, many of these studies have only managed to recapitulate a fraction of the total resistance levels observed in the field [10,11].

An additional factor in IRM strategies are the evolutionary fitness costs imposed by resistance alleles. Alleles that pose high costs will tend to revert back to their susceptible form once the selective pressure (pesticide) is removed. Variants that do not pose such a cost can persist indefinitely. Fitness costs related to drug resistance have long been the subject of investigation in clinical settings with resistant bacteria and cancer lines [12,13], and have also been subjected to investigation in insects [14]. From these studies it has become clear that the severity, and indeed presence, of a fitness cost brought about by a given resistance allele depends on the particular allele and the genetic background in which it is observed. However, the understanding of the costs of each variant and their epistatic effects are poorly understood.

Of particular interest to the insecticide community are the mechanisms underpinning pyrethroid resistance and their resulting evolutionary implications. This large class of structurally related insecticides targets the voltage-gated sodium channel (the orthologues to the *para* gene in *Drosophila*), and representative pyrethroids such as deltamethrin have been widely used in both agricultural and public health related pest control since the 1970s. Widespread use was followed by the appearance of several independent resistance mechanisms. Among several examples, the P450 *Cyp6BQ23* was found to be overexpressed in the pollen beetle *Brassicogethes aeneus* [15], while other resistant strains carried the *kdr* (L1014F) substitution in *para* [16]. In the mosquito *Aedes aegypti*, the main vector of yellow fever worldwide, a similar array of mechanisms have been identified including the overexpression of *Cyp9J28*; [17] and another mutation in *para* (V1016G). However, the interaction of these alleles *in vivo* has not, to our knowledge, been studied, neither in terms of contribution to pyrethroid resistance nor any resulting fitness cost.

Here, we report the generation of transgenic *Drosophila melanogaster* lines expressing pyrethroid metabolizing cytochrome P450 enzymes from major mosquito vectors (*A. aegypti*) and agricultural pests (*B. aeneus*) in a genetic background where we have engineered by CRISPR/Cas9 specific homozygous target-site resistance mutations in the voltage-gated sodium channel (*para*), also found in these insects. This strategy enabled us to directly measure the resulting resistance phenotypes, encountering either the contribution of both mechanisms or each one separately, with very limited confounding genetic effects.

# 2. Material and methods

## (a) *Drosophila* strains

*Drosophila* strains used in this study are shown in the electronic supplementary material, table S1. Strain *yw nos int;* attP40 [18] was a gift by Pawel Piwko and Christos Delidakis (IMBB/FORTH). CRISPR/Cas9 genome modification was performed at strain y1 M{nos-Cas9.P}ZH-2A w*, where Cas9 is expressed under the control of *nanos* promoter [19] (herein referred as nos.Cas9, #54591 in the Bloomington *Drosophila* stock centre). Background strain *yellow white* (*yw*) and several balancer lines (see the electronic supplementary material, table S1) are part of the IMBB/FORTH facility fly collection (kindly provided by Prof. Christos Delidakis, IMBB and University of Crete). The HR-GAL4 driver line is previously described [20], while the responder line UAS.AaegCYP9J28 (herein referred as UAS-CYP9J28) was generated also as described previously [17]. All flies were kept at a temperature of 25°C, humidity 60–70% and 12 : 12 h photoperiod on a standard fly diet.

## (b) Amplification and sequencing of *para* target regions

DNA from nos.Cas9 *Drosophila* adults was extracted with DNAzol (MRC, Cincinnati, OH) following the manufacturer's instructions. Several primers (paraInF, paraInR, kdrF, kdrR, exoF, exoR, electronic supplementary material, table S2) were designed based on the *para* gene sequence in order to amplify and sequence overlapping fragments that correspond to a 2585 bp sequenced genomic region of strain nos.Cas9 (X: 16485234:16487819, numbering according to BDGP6.22 genome assembly) which contains the exons that harbour positions L1014 and V1016, respectively. The amplification reactions were performed using KapaTaq DNA polymerase (Kapa Biosystems, Wilmington, MA). The conditions were 95°C for 2 min for initial denaturation followed by 30–35 cycles of denaturation at 95°C for 30 s, annealing at 54–60°C for 15 s, extension at 72°C for 90 s and a final extension step for 2 min. The polymerase chain reaction (PCR) products were purified with a PCR clean-up kit (Macherey-Nagel, Düren, Germany) according to manufacturer's instructions. Sequencing of the products was performed from both ends at Macrogen (Amsterdam, The Netherlands).

## (c) Strategy for genome editing

An ad hoc CRISPR–Cas9 strategy was implemented in order to generate *Drosophila* strains bearing either mutation (equivalent to L1014F and V1016G according to housefly *Musca domestica* numbering) in the *para* gene. Based on the genomic sequence of *para* obtained for strain nos.Cas9 several CRISPR targets in the desired region were identified using the Optimal Target Finder online tool [21] (http://targetfinder.flycrispr.neuro.brown.edu/). We selected three CRISPR targets in total, which had minimal predicted off-target effects. Targets para935 and para406 were used to obtain L1014F, while targets para406 and para205 were used for V1016G (figure 1). In order to generate single guide RNAs (sgRNAs) targeting those sequences, three different RNA-expressing plasmids were generated (sgRNA935, sgRNA406 and sgRNA205, respectively) based on the vector pU6-BbsI chiRNA [22] following digestion with BbsI and ligation of three relevant double-stranded oligonucleotides, which were generated by annealing single stranded oligonucleotides 935F/935R, 406F/406R and 205F/205R, respectively (electronic supplementary material, table S2). Following ligation and transformation, single colonies for each construct were picked and checked for the correct insert by performing colony PCR using T7 universal primer and the reverse

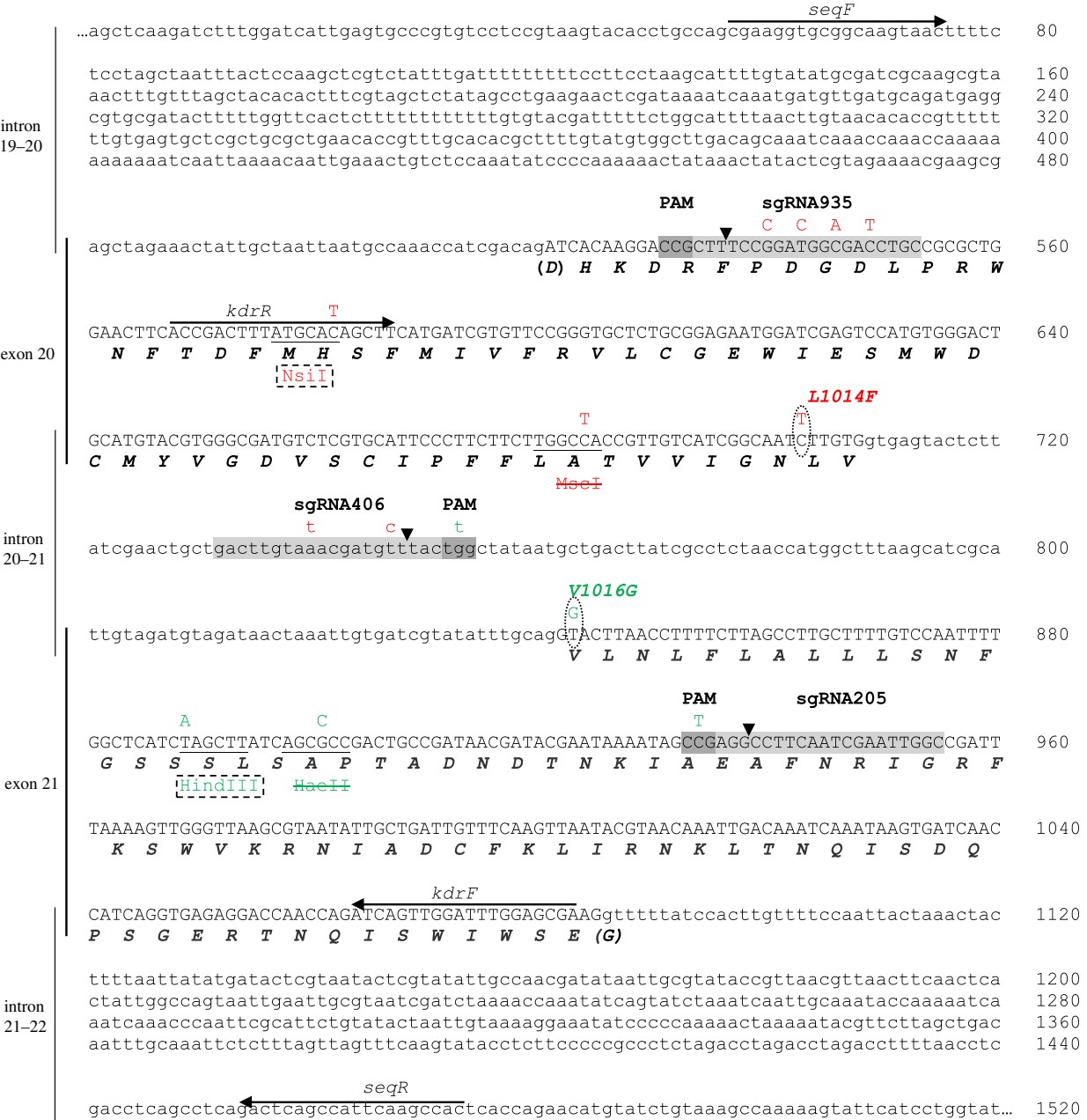

**Figure 1.** CRISPR/Cas9 strategies for the generation of genome modified flies bearing mutations L1014F and V1016G. Nucleotide and deduced amino acid sequence of a 1520 bp fragment of *para*, encompassing exons 20 and 21 that contain positions 1014 and 1016, respectively (*M. domestica* numbering) of the *Drosophila melanogaster* amino acid sequence. Light grey areas indicate the CRISPR/Cas9 targets selected (sgRNA935, sgRNA406 and sgRNA2015), while dark grey areas indicate the corresponding PAM (-NGG) triplets. Vertical arrows denote break points for CRISPR/Cas9-induced double-stranded breaks. Red lettering indicates the differences introduced in exon 20 for the generation of L1014F, while green lettering indicates the differences introduced in exon 21 for the generation of V1016G. Ovals mark non-synonymous differences between the target (wild-type) and donor (genome modified) sequences. Synonymous mutations incorporated for diagnostic purposes, as well as to avoid cleavage of the donor plasmid by the CRISPR/Cas9 machinery, are shown above the nucleotide sequence. Restriction sites abolished because of the genome modification are shown with strikethrough letters and the corresponding sequence is underlined. Restriction sites introduced because of the genome modification are shown in dashed boxes and the corresponding sequence is also underlined. Horizontal arrows indicate the positions of primer pairs kdrF/kdrR and seqF/seqR (electronic supplementary material, table S2) used for sequencing of the genome modified alleles. (Online version in colour.)

primer for each double stranded DNA. The sequence of each sgRNA expressing plasmid was verified by sequencing (Macrogen).

Two different donor plasmids, vgscL1014F and vgscV1016G were synthesized de novo (Genscript, Piscataway, NJ) to facilitate homologous directed repair for the generation of strains L1014F and V1016G, respectively (newly synthesized sequences were subcloned in the pUC57 vector EcoRV site; relevant insert sequences for each donor plasmid are shown in the electronic supplementary material, figure S1). Each plasmid contained two approximately 900 bp homology arms flanking the target region between sgRNA targets para935 and para406 (for L1014F) or para406 and para205 (for V1016G) (figure 1). The

target regions were specifically designed in order to contain the desired mutations along with certain additional synonymous mutations (figure 1) serving either as molecular markers (to facilitate molecular screening of CRISPR events), or to prevent unwanted CRISPR digestion of the donor itself.

### (d) Molecular screening and establishment of genome modified lines

Injection of nos.Cas9 pre-blastoderm embryos was performed at the IMBB/FORTH facility with injection mixes containing

75 ng μl$^{-1}$ of each sgRNA plasmid vector and 100 ng μl$^{-1}$ of donor template as previously described [23]. Hatched larvae were transferred into standard fly artificial diet and after 9–13 days $G_0$ (generation zero) surviving adults were collected and individually backcrossed with nos.Cas9 flies. In order to screen for CRISPR events, $G_1$ progeny from each cross were pooled into batches of approximately 30 and genomic DNA extraction was performed en masse in order to be screened with two different strategies. Initially, 2 μg of pooled genomic DNA (gDNA) were digested with MscI (for L1014F crosses) or HaeII (for V1016G crosses); these enzymes cut only the wild-type alleles but not potential mutant alleles in each DNA pool. Then, the strategy for screening for L1014F mutants consists of amplification with specific primers 1014UP/1014DOWN (electronic supplementary material, table S2) that were designed taking into account the synonymous mutations introduced in the relevant target sequences in donor template vgscL1014F, in order to generate a 234 bp diagnostic fragment that is specific to genome modified alleles, but not wild-type ones (electronic supplementary material, figure S2A). PCR was performed with Kapa Taq polymerase as previously described using approximately 60 ng of digested template DNA mix. For screening of V1016G mutants, an alternative strategy was used, which consists of PCR amplification with the 'generic' primer pair kdrF/kdrR (electronic supplementary material, table S1) which was designed in order to amplify a 516 bp fragment that may be derived by either wild-type (if still present, given the initial enzymatic cleavage of the template DNA mix) or genome modified alleles. Following PCR amplification, the product was digested with the diagnostic enzyme HindIII introduced in the vgscV1016G donor plasmid sequence, producing two diagnostic fragments of 324 and 192 bp (electronic supplementary material, figure S2B). Crosses that proved positive for genome modified alleles were further explored in order to identify individual flies bearing mutant alleles and establish homozygous lines. DNA was extracted from several homozygous female and hemizygous male adults, amplified by using primers kdrF/kdrR or seqF/ seqR (electronic supplementary material, table S2) and the relevant amplification fragments were sequence verified (Macrogen) for the presence of the desired mutations (electronic supplementary material, figure S2C).

## (e) Generation of transgenic *Drosophila* expressing *Cyp6BQ23*

In order to generate a transgenic *D. melanogaster* strain conditionally expressing *Cyp6BQ23*, a GAL4/UAS strategy was employed. The responder strain, UAS-CYP6BQ23, was generated by PhiC31 integrase mediated attB insertion at an attP40 landing site [24]. An ad hoc integration vector, dPelican-attB-UAS_ CYP6BQ23 was generated by replacing the insert of plasmid dPelican-attB-UAS_CYP6A51 we had previously generated [25]. We performed de novo synthesis (Genescript) of the CYP6BQ23 coding sequence (GenBank acc. no. KC840055.1) with some modifications in order to optimize for expression in *Drosophila*, i.e. introducing a CACC Kozac-consensus sequence just upstream of the initiation codon and taking into account codon usage optimal for *Drosophila* (full construct sequence shown in the electronic supplementary material, figure S1). An MluI/XhoI fragment encompassing the CYP6BQ23 coding sequence was subcloned into dPelican-attB-UAS_CYP6A51 [25] plasmid backbone that had been digested with MluI and XhoI so that the existing CYP6A51 expression cassette was removed and replaced by the CYP6BQ23 fragment downstream of 5xUAS and just upstream of an SV40 polyadenylation sequence, to produce the final recombinant plasmid dPelican.attB.UAS_CYP6BQ23. This plasmid also contains a mini-*white* marker gene for *Drosophila*. The sequence was verified using sequencing primers pPel_uas F and pPel_sv40 R [26] and the recombinant plasmid was used to

inject pre-blastoderm embryos of the *D. melanogaster* strain *yw nos int*; attP40. Injected $G_0$ flies were outcrossed with *yw* background flies and $G_1$ progeny was screened for $w^+$ phenotypes (red eyes) indicating integration of the recombinant plasmid. Independent transformed lines were crossed with a strain bearing a balancer for the second chromosome (*yw; CyO/Sco*), and $G_2$ flies with red eyes and *Cy* phenotype were selected and crossed among themselves to generate homozygous UAS-CYP6BQ23 flies used to establish the transgenic responder line population.

## (f) Generation of null background strain *yw;attP40*

In order to generate a *Drosophila* line that is fully equivalent to the UAS-CYP6BQ23 strain in terms of genetic background and can be used as null control in downstream experiments, male non-injected flies of the *D. melanogaster* strain *yw nos int*; attP40 were outcrossed with female *yw* background flies, and male $G_1$ progeny (not carrying the *yw nos int* chromosome) was crossed with females bearing a balancer for the second chromosome (*yw; CyO/Sco*). $G_2$ flies with *Cy* phenotype were selected and crossed among themselves to generate homozygous *yw; attP40* flies that have essentially the same genetic background with the transgenic responder line population, apart from the UAS-CYP6BQ23 expression cassette.

## (g) Generation of homozygous recombinant *yw; HR-GAL4>UAS-CYP9J28(2N)* strain

We generated a strain bearing both HR-GAL4 and UAS-CYP9J28 in the second chromosome by genetic recombination, as shown in the electronic supplementary material, figure S3. This was performed via a cross between lines HR-GAL4 [20] and UAS-CYP9J28 [17] that produces a heterozygous genotype (*y*)*w*;HR-GAL4>UAS-CYP9J28. However, while this genotype produces a detectable resistant phenotype in contact bioassays [17], preliminary topical application bioassays indicated that only marginal (i.e. not always significant) changes in resistance were detected (data not shown). Thus, heterozygous (*y*)*w*;HR-GAL4>UAS-CYP9J28 females (where chromosomal crossover is feasible) were crossed to *yw*; *CyO/Sco* balancer flies and the progeny screened for genetic recombination events as shown in the electronic supplementary material, figure S3. We identified heterozygous *yw*; [HR-GAL4_UAS-CYP9J28]/CyO recombinant flies bearing both HR-GAL4 and UAS-CYP9J28 transgenes in the second chromosome and these were intercrossed to generate the homozygous *yw*; HR-GAL4>UAS-CYP9J28(2N) strain, which contains two copies of both driver and responder transgenes.

## (h) Generation of driver and responder lines in genome modified (mutant) background

Taking into account that the voltage-gated sodium channel (*para*) gene in *Drosophila* is located at the X chromosome, while the HR-GAL4 driver strain used in our laboratory, the *Cyp9J28* insertion site, as well as the attp40 insertion site bearing the *Cyp6BQ23* transgene, are all located in the second chromosome, we devised a simple genetic crosses strategy (electronic supplementary material, figure S4) to: (i) introduce both the HR-GAL4 driver and the UAS-CYP6BQ23 responder transgene in a *para*$^{L1014F}$ genetic background in order to generate strain *para*$^{L1014F}$; HR-GAL4> UAS-CYP6BQ23 that represents the 'beetle' allele combination, and (ii) generate a strain bearing the linked [HR-GAL4_UAS-CYP9J28] chromosome 2 (see (g) above) in a *para*$^{V1016G}$ genetic background (strain *para*$^{V1016G}$;HR-GAL4>UAS-CYP9J28(2N)) that represents the 'mosquito' allelic combination. As shown in the electronic supplementary material, figure S4, the crossing scheme results in lines where the X chromosome is derived from nos.Cas9 strain, the second chromosome from the

respective transgenic line (HR-GAL4, *yw nos int; attP40* or UAS-CYP9J28) and the other chromosomes from *yw* (note that all balancer lines as well as nos.Cas9, *yw nos int; attP40* and UAS-CYP9J28 had been originally generated in *yw* or extensively outcrossed to it in the IMBB fly facility).

### (i) GAL4/UAS-expression in *Drosophila melanogaster*

In order to drive conditional expression of *Cyp9J28* or *Cyp6BQ23* in wild-type or mutant genetic backgrounds we used the HR-GAL4 driver [20] which drives expression in specific tissues related to detoxification (malpighian tubules, midgut and fat body). Transgenic responder virgin females were crossed with HR-GAL4 males and the progeny was used in toxicity bioassays with deltamethrin in order to validate the potential of each line to confer insecticide resistance. Crosses of *yw; attP40* or nos.Cas9 virgin females with HR-GAL4 males were used as negative controls.

### (j) Extraction of RNA, complementary DNA synthesis, reverse transcription polymerase chain reaction and quantitative polymerase chain reaction

Reverse transcription PCR was performed in order to confirm expression of *Cyp6bq23* or *Cyp9j28* in the progeny. Total RNA was extracted from pools of 20 adult *Drosophila* flies (1–3 day old) using Trizol reagent (MRC), according to the manufacturer's instructions. Extracted RNA samples were treated with Turbo DNase (Ambion, Foster City, CA) to remove genomic DNA and 2 µg of treated RNA was used to generate first strand complementary DNA (cDNA) using oligo-dT$_{20}$ primers with Superscript III reverse transcriptase (Invitrogen, Carlsbad, CA). One microliter of cDNA was used in the PCR reaction using specific primers for each transgene and for *rpl11* (ribosomal protein L11) which served as a reference gene (electronic supplementary material, table S1). The conditions of the reactions were 95°C for 5 min followed by 35 cycles of 95°C for 30 s, 55°C for 30 s, 72°C for 30 s and final extension for 2 min.

A two-step quantitative-reverse transcriptase PCR was performed in order to analyse the expression levels of *Cyp6BQ23* and *Cyp9j28* between genotypes bearing the relevant expression trangenes in either wild-type or genome modified *para* background (figure 2*c*,*d*). Quantitative PCR (qPCR) was conducted using the Kapa SYBR Fast qPCR Master Mix kit (Kapa Biosystems) and the reactions were carried out in the Bio-Rad CFX Connect using the following conditions: 95°C for 2 min, followed by 40 cycles at 95°C for 5 s and 60°C for 30 s. The efficiency of the qPCR reaction for each primer pair was assessed in 10-fold dilution series of pooled cDNA samples. The experiment was performed using three biological and two technical replicates. Relative expression was normalized to the reference genes *rpl11* and *rpl32*, while the analysis was conducted as previously described [27]. All primer sequences are shown in the electronic supplementary material, table S2.

### (k) Toxicity bioassays

Deltamethrin (99.6%) of technical grade was purchased from Sigma Aldrich (Darmstadt, Germany) and used in contact assays and topical applications. Contact assays were performed as described previously [28]. More specifically, 20 adult female flies (1–3 days old, as per [20,28]) were used for each toxicity assay. Flies were collected in plastic vials and transferred in scintillation vials coated with insecticide. Serial dilutions of 6–7 concentrations of technical grade deltamethrin in acetone were used for dose response bioassays, while vials coated only with acetone served as control. The vials were plugged with cotton that was kept moist with 5% sucrose solution. Each deltamethrin concentration was assayed in three replicates. Knockdown was scored for 180 min with 15–30 min intervals and mortality was scored after 24 h. Topical application of deltamethrin was performed on 1–3 day old female flies. Deltamethrin was dissolved in acetone and serial dilutions were made to generate the appropriate concentrations. Each insecticide concentration (or acetone as negative control) was applied in a dose of 1 µl per fly using a 10 µl Hamilton syringe. Flies were immobilized by keeping them on an ice cold slide. For each concentration 40 flies were tested. Following insecticide application the flies were transferred into glass scintillation vials covered with cotton moisturized with 5% sucrose solution. The vials were maintained in a 25°C incubator while mortality was scored after 24 h.

### (l) Life table parameters

Determination of life table parameters was performed as previously described [23]. To determine developmental time and sex ratio, cages with 50 virgins (1–3 day old) and 20 males were capped with cherry juice-agar plates supplemented with yeast, left to cross overnight and after plate replacement the flies were left to lay eggs for 4–5 h. Eggs were transferred into vials with standard fly food in batches of *ca* 50 (10 replicates for each genotype). Pupation, adult emergence time and total number of males/females were scored daily from day 8. To monitor daily and total fecundity, c. 15 females from each genotype were mated, transferred in small cages capped with 35 mm yeast-supplemented cherry agar plates, and the number of eggs laid by each female was counted daily.

### (m) Statistical analyses

Concentration-response data of each bioassay setup were collected and analysed with ProBit analysis using POLOPLUS (LeOra Software, Berkeley, CA) in order to calculate lethal concentrations of the 50% of the population subjected to the experiment (LC$_{50}$ values), 95% fiducial limits (FL), linearity of the dose-mortality response and statistical significance of the results. A $\chi^2$-test was used to assess how well the individual LC$_{50}$ values observed in the bioassays agree with the calculated linear regression lines. The LC$_{50}$ values and resistance ratio (RR) are considered significant if the 95% FL did not include 1 [29]. Life table parameter data (electronic supplementary material, dataset 1) were analysed for significant differences between strains with one-way ANOVA using the software GRAPHPAD PRISM 8.0.2. A two-tailed unpaired student's *t*-test (also using GRAPHPAD PRISM) was carried out in order to compare relative expression in qPCR data.

## 3. Results

### (a) Generation of *Drosophila* lines expressing detoxification enzymes in a genetic background bearing target-site resistance mutations in *para*

We used CRISPR/Cas9 genome engineering (figure 1) to generate strains bearing homozygous target-site resistance mutations in the voltage-gated sodium channel (*para*) of *D. melanogaster* (strain *para*[L1014F] bearing mutation L1014F (*kdr*) and strain *para*[V1016G] bearing mutation V1016G). Additionally, we employed GAL4/UAS for the expression of known detoxification enzyme CYP6BQ23 from the pollen beetle *B. aeneus*. We also used a previously generated strain for expression of CYP9J28 from the mosquito *A. aegypti* under the control of HR-GAL4 driver. We used genetic recombination to bring HR-GAL4 and UAS-CYP9J28 in the same chromosome, resulting in a strain bearing two copies of each transgene (*yw*; HR-GAL4>UAS-CYP9J28(2N)) in

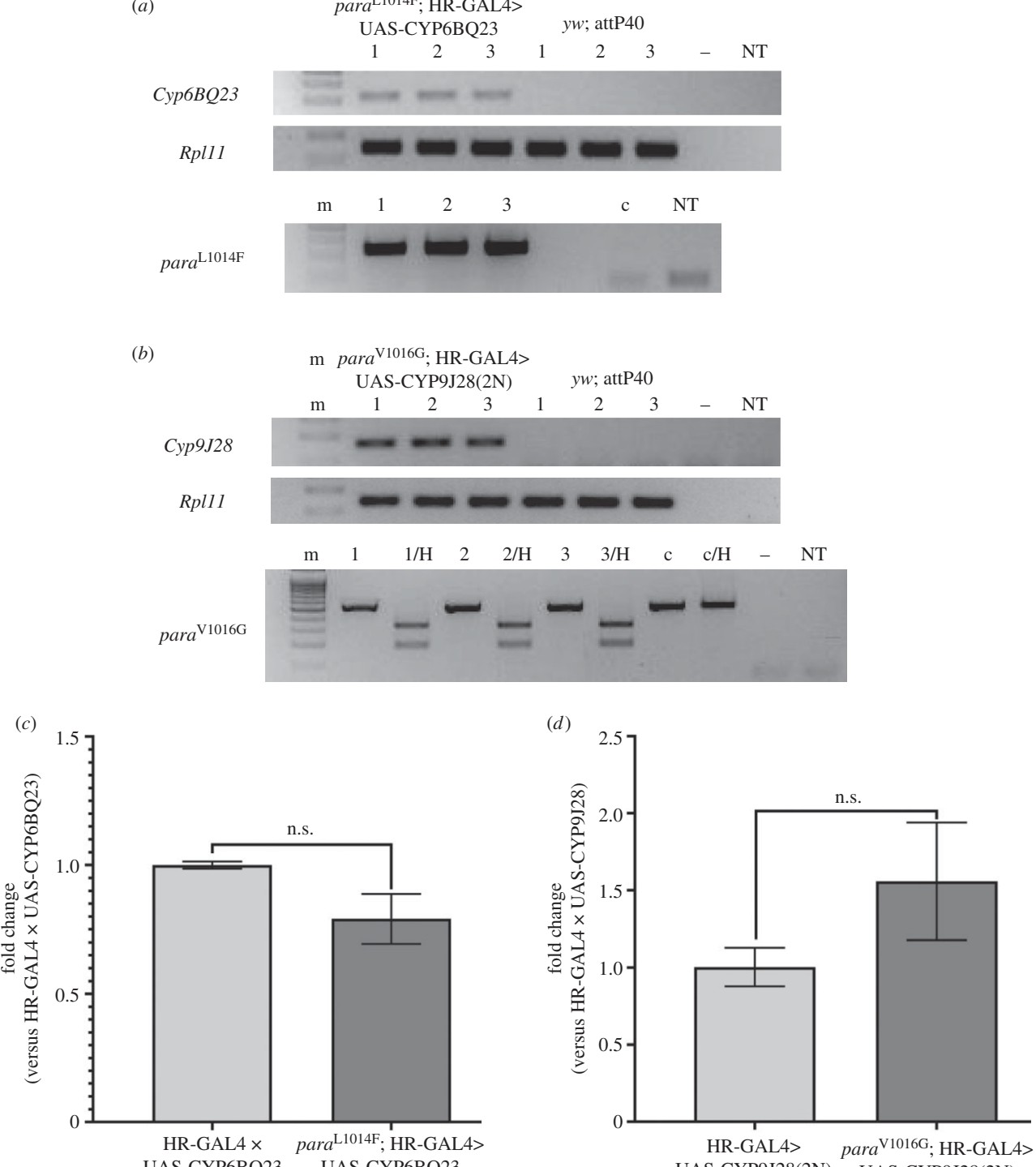

**Figure 2.** P450 overexpression in the flies bearing the *kdr* mutations. (*a*) (top) *Cyp6bg23* expression is confirmed by reverse transcription and PCR amplification of cDNAs. Lanes *para*[L1014F];HR-GAL4>UAS-CYP6BQ23 (1–3) indicate three biological replicates of the flies tested for the overexpression of the transgene. Lanes *yw*; attP40 (1–3) indicate the three biological replicates of the control line. The same cDNAs were used to amplify the housekeeping gene *rpl11* as a reference gene. —: no reverse transcription control (to monitor for genomic DNA contamination); NT: no template control. (bottom): The presence of L1014F mutation in the same flies is tested by PCR of genomic DNA with allele-specific primers. c: *yw*; attP40 negative control DNA. (*b*) (top) *Cyp9j28* expression is similarly confirmed. Lanes *para*[V1016G];HR-GAL4>UAS-CYP9J28(2N) (1–3) indicate three biological replicates of the flies tested for the overexpression of the transgene, while lanes *yw*; attP40 (1–3) indicate the three biological replicates of the control line. The same cDNAs were used to amplify the housekeeping gene *rpl11* as a reference gene. —: no reverse transcription; NT: no template. (bottom): The presence of V1016G mutation in the same flies is tested by PCR of genomic DNA with generic primers and subsequent digestion of the product with HindIII (/H). c: *yw*; attP40 negative control DNA. (*c,d*) qRT-PCR for evaluation of P450 expression levels in different strains. The Ct values of strains expressing CYP6BQ23 (*c*) and CYP9J28 (*d*) were calculated in the absence or presence of the relevant *para* mutations. No significant difference in expression was observed ($p = 0.0618$ for CYP6BQ23, $p = 0.1161$ for CYP9J28).

order to obtain higher expression levels providing a readily detectable effect in topical application assays (see below). Standard genetic crosses enabled *Cyp6BQ23* transgenic expression in *para*[L1014F] genetic background and *Cyp9J28* transgenic expression in *para*[V1016G] background so that both mechanisms were combined (figure 2).

**Table 1.** Topical application deltamethrin bioassay responses of transgenic flies expressing pyrethroid metabolizing P450s alone or along engineered target-site resistance mutations in their voltage-gated sodium channel (*para*).

| strain/cross | LD$_{50}$ (ng fly$^{-1}$) | (95% FL) | slope (±s.e.) | RR |
|---|---|---|---|---|
| HR-GAL4 × *yw*; attP40 | 3.10 | (2.65–3.65) | 3.59 (±0.47) | 1 |
| HR-GAL4 × UAS-CYP6BQ23 | 17.8 | (12.50–21.65) | 4.27 (±0.87) | 5.74 |
| *yw*;HR-GAL4>UAS-CYP9J28(2N)[a] | 5.49 | (4.051–6.60) | 4.4 (±1.09) | 1.77 |
| nos.Cas9 | 3.33 | (1.3–5.1) | 2.259 (±0.39) | 1.07 |
| *para*$^{L1014F}$ | 39.49 | (23.1–53.95) | 2.949 (±0.39) | 12.74 |
| *para*$^{V1016G}$ | 9.30 | (4.98–14.55) | 1.696 (±0.36) | 3.00 |
| *para*$^{L1014F}$; HR-GAL4>UAS-CYP6BQ23[b] | 233.08 | (161.70–333.85) | 1.508 (±0.21) | 75.19 |
| *para*$^{V1016G}$; HR-GAL4>UAS-CYP9J28(2N)[c] | 61.53 | (47.48–78.50) | 4.851 (±0.80) | 19.85 |

[a]homozygous recombinant *yw*; HR-GAL4>UAS-CYPJ28(2N) contains two copies of driver and responder.
[b]*para*$^{L1014F}$; HR-GAL4 × *para*$^{L1014F}$; UAS-CYP6BQ23.
[c]*para*$^{V1016G}$; HR-GAL4>UAS-CYPJ28(2N) contains two copies of driver and responder in *para*$^{V1016G}$ X-chromosome background.

## (b) Toxicity bioassays in *Drosophila* indicate synergistic action of different resistance mechanisms

Slow uptake contact bioassay experiments (all results shown in the electronic supplementary material, table S3) showed that resistance levels were extremely high in *para*$^{L1014F}$;HR-GAL4>UAS-CYP6BQ23 flies (totally insensitive at deltamethrin doses exceeding 5000 µg vial$^{-1}$), compared to the control flies nos.Cas9 (LC$_{50}$ 5.45 (2.40–8.57) µg vial$^{-1}$) or the flies bearing any of the resistant mechanisms alone (RR$_{L1014F}$: 158.9; RR$_{Cyp6BQ23}$: 9.68 compared to relevant control flies nos.Cas9 and HR-GAL4 × *yw*; attP40, respectively).

Thus, in order to more precisely quantify the intensity of resistance phenotype, topical application bioassays were used. The results are shown in table 1. *para*$^{L1014F}$ flies exhibit 12.74-fold resistance to deltamethrin, while flies expressing *Cyp6BQ23* in wild-type *para* background exhibit 5.74-fold resistance compared to controls (nos.Cas9 and HR-GAL4 × *yw*; attP40), which had an absolutely similar response to deltamethrin toxicity.

Flies expressing *Cyp6BQ23* in *para*$^{L1014F}$ background displayed an almost multiplicative RR compared to the control (RR$_{combined}$: 75.19 ≥ RR$_{Cyp6BQ23}$: 5.74 × RR$_{L1014F}$: 12.74). In the case of the resistance alleles known from *A. aegypti* an even more striking effect was found: *para*$^{V1016G}$ flies show modest levels of resistance (RR: 3.00), while flies stably expressing *Cyp9J28* following genetic recombination between the UAS-CYP9J28 responder with the HR-GAL4 driver (bearing two copies of each) also exhibit modest resistance (RR: 1.77) compared to controls. However, the *para*$^{V1016G}$;HR-GAL4>UAS-CYP9J28(2N) flies displayed a significantly greater RR than the product of the individual RRs (RR$_{combined}$: 19.85 ≫ RR$_{Cyp9J28}$: 1.77 × RR$_{V1016G}$: 3.00).

## (c) The presence of multiple resistance alleles may be associated with some fitness disadvantage

By contrast to the lines bearing only one resistant mechanism, which (with a possible exception on *para*$^{L1014F}$ fecundity) exhibited no statistically significant difference compared to the control lines (electronic supplementary material, dataset 1), both the 'super-resistant' lines, *para*$^{L1014F}$;HR-GAL4>UAS-CYP6BQ23 and *para*$^{V1016G}$;HR-GAL4>UAS-CYP9J28(2N) showed a significant cost in development time as indicated by one-way ANOVA of pupation time (figure 3 and electronic supplementary material, table S4, pupation after 7–8 days). For the 'beetle' allelic combination, *para*$^{L1014F}$;HR-GAL4>UAS-CYP6BQ23 flies exhibit some developmental delay compared to nos.Cas9 controls (p$_{d7}$ = 0.0249), as well as to *para*$^{L1014F}$ (p$_{d7}$ = 0.0044 and p$_{d8}$ = 0.0274). Regarding the 'mosquito' combination, this is also evident in comparisons of *para*$^{V1016G}$;HR-GAL4>UAS-CYP9J28(2N) flies against nos.Cas9 (p$_{d7}$ = 0.0039, p$_{d8}$ = 0.0045), *para*$^{V1016G}$(p$_{d7}$ = 0.0110, p$_{d8}$ = 0.0048), *yw*;HR-GAL4>UAS-CYP9J28 (p$_{d7}$ = 0.0004, p$_{d8}$ < 0.0001) and HR-GAL4>*yw*;attP40 (p$_{d7}$ = 0.0093, p$_{d8}$ = 0.0089). Furthermore, *para*$^{V1016G}$;HR-GAL4>UAS-CYP9J28(2N) flies also exhibit a significant cost in total fecundity after 20 days, compared to all controls tested (figure 3 and electronic supplementary material, table S5).

## 4. Discussion

An old enigma in insect toxicology, the putative synergistic action of target site resistance mutations and upregulated cytochrome P450s in pyrethroid resistance of major disease vectors and agricultural pests, has been functionally resolved. Specifically, field evolved P450s conferring pyrethroid resistance and target site resistance alleles were individually introduced in a single *in vivo* system. Genotypes overexpressing P450s in addition to the target-site resistance displayed a multiplicative RR equal or greater than the product of the RRs obtained for the individual resistance mechanisms. This is in line with previous studies focusing on synergism associated with insecticide resistance alleles (reviewed in [6]), which found that the combination of resistance alleles was mostly multiplicative.

While the work presented here then largely agrees with prior estimations of P450 target site synergism, the use of *D. melanogaster* in this study provides several advantages. Most critically *Drosophila* reduces confounding genetic factors arising from different backgrounds which are known to cause substantial variation in 'wild-type' lines [30]. While the flies used in this study were not completely isogenic, the backgrounds were much more similar compared to previous studies considering synergism in model organisms [6] and

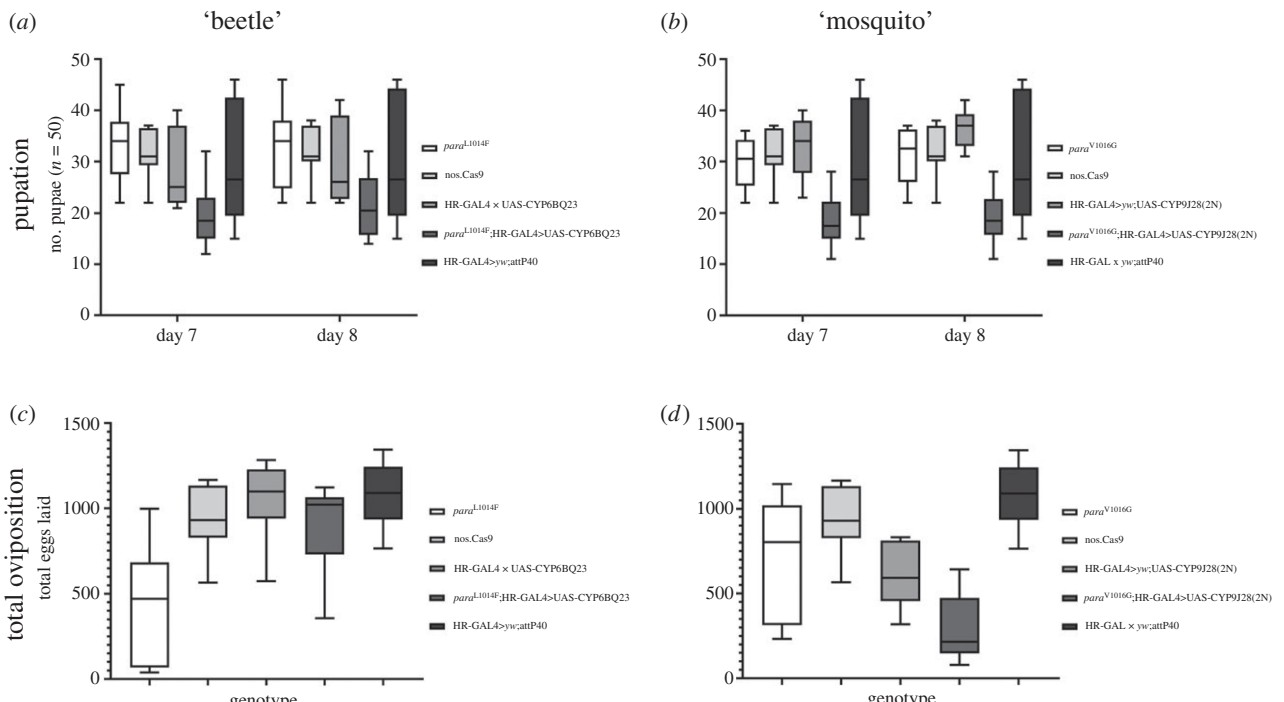

**Figure 3.** Evaluation of 'super-resistant' fly lines fitness. (*a,b*): one-way ANOVA of pupation after 7–8 days in strains bearing both resistance alleles for the 'beetle' (*a*) and 'mosquito' (*b*) allele combinations compared to controls, indicating an extended developmental time when both alleles are present. All *p*-values are shown in the electronic supplementary material, table S4. (*c,d*): one-way ANOVA of total oviposition in strains bearing both resistance alleles for the 'beetle' (*c*) and 'mosquito' (*d*) allele combinations. Oviposition is significantly reduced in $para^{V1016G}$; HR-GAL4>UAS-CYP9J28(2N) flies against all controls, indicating reduced overall fecundity. All *p*-values are shown in the electronic supplementary material, table S5. For the full life table parameters dataset, see the electronic supplementary material, dataset 1.

provide similar levels of resistance in topical application bioassays (table 1). When genetic background was controlled for in pest species, it has only been done by backcrossing individual variants such as the 1016 mutation in *A. aegypti* [31]. *Drosophila* provides an easier alternative, but caution must be taken between interpreting findings across species, taking into account not only the powers but also the limitations of this system (for a detailed relevant discussion see [32,33]).

The mechanism for the apparent synergism between P450s and target site mutations is still not fully understood. It may involve only the parent compound, i.e. reduced binding affinity for the target site could simply give P450s additional time to perform insecticide metabolism and avoid saturation. This is thought to be the case with P-glycoprotein CYP3A4 synergism in humans, whereby the former increases the time a compound spends in the intestine and thereby increases its chance of being metabolized by the latter [34]. The higher $RR_{combined}$ in the slow uptake compared to the fast-uptake topical application seems to corroborate this hypothesis (electronic supplementary material, table S3 and table 1, respectively). Alternatively, the less toxic P450-generated metabolites might bind the mutant target receptors disproportionately less effectively, thus manifesting in a synergistic phenotype. It is currently unknown what, if any, affinity the P450 derived metabolites of deltamethrin would have for *para*, but it has been suggested that their accumulation is likely to be detrimental in mosquitoes and that their metabolism by certain P450s, such as the *Anopheles gambiae* CYPZ8 and CYP6Z2, is a secondary but important mechanism of insecticide resistance [35]. In addition, examples from other compounds may provide some guidance. Imidacloprid is 'detoxified' by P450s into a variety of metabolites which still show levels of toxicity

that would probably be relevant *in vivo* [36,37]. Nevertheless, there is currently no data regarding the binding of various pyrethroid metabolites for different alleles of *para*, so such a hypothesis awaits functional validation.

In addition to differences in resistance level, fitness costs were also observed for the various genotypes used in this study. While genotypes carrying a single resistance allele behaved similarly to the control lines (with a possible exception regarding $para^{L1014F}$ fecundity), the combination of these alleles significantly increased the developmental time both in the 'beetle' and 'mosquito' genotype combinations and significantly reduced fecundity in the 'mosquito' combination against multiple controls contributing to the genetic background (figure 3). It thus seems possible that these alleles, may exert a fitness cost only in certain backgrounds, an implication which has far reaching ramifications for IRM. This hypothesis is also supported by studies done directly on these pest species. The combination of high level *Cyp6BQ23* expression and *kdr* mutations is extremely rare in pollen beetle populations, supporting the fitness cost theory [16], but the mechanism underpinning this phenomenon is not known. By contrast, the results presented here on the V1016G mutation contradict those obtained by backcrossing the mutation into a susceptible *A. aegypti* background, although the precise mutation in that study was different [31]. These data collectively suggest epistatic effects between different resistance mechanisms and highlight the need for fitness cost assessment to be done in multiple backgrounds. Further work will thus be needed to establish and characterize the evolutionary significance of these resistance alleles in the field.

Several groups are currently developing and applying DNA-based technologies for insecticide resistance monitoring [8,9]. Our study shows that these molecular diagnostics need

careful calibration, integration and interpretation, as an apparent epistasis (i.e. different phenotype depending on the genetic background) is present, and thus they may or may not diagnose the importance of resistance in the field. The reconstruction of complex resistance phenotypes by reverse-genetics based simultaneous introduction of individual mechanisms in a susceptible genetic background enhances our ability to elucidate the contribution of each individual molecular mechanism in the resistance phenotype; a concept that is perhaps best represented by the famous quote found at the blackboard of R. P. Feynman (1918–1988) at the time of his death ('*What I cannot create, I do not understand*'). Although significant research effort remains to be done, the present study provides a 'proof of principle' of the applicability of such a reconstructed resistance 'network of interactions' within a model *Drosophila* 'test tube' that enables the validation of hypotheses regarding the molecular mechanisms contributing to insecticide resistance phenotypes in field populations; in other words, to (re)create the interactions among different mechanisms, so that we can gain insight on their specific role.

**Data accessibility.** All data generated or analysed during this study are available in this manuscript and its electronic supplementary material.

**Authors' contributions.** V.D. and J.V. conceived the project and designed the experiments; G.-R.S., R.P. V.D., S.K. and I.C. generated fly lines; G.-R.S. and R.P. performed toxicity bioassays and qPCR; G.-R.S. performed fitness cost experiments; V.D., S.D., R.N. and J.V. analysed the data, contributed to scientific discussion and wrote the manuscript, which has been revised and approved by all authors.

**Competing interests.** The authors declare no competing interests.

**Funding.** The research has been supported by the European Union Horizon 2020 Framework Programme (688207-SuperPests) grant to J.V., by a Fontation Santé research grant to V.D., by the Hellenic Foundation for Research and Innovation (HFRI) under the HFRI PhD fellowship grant (fellowship no. 10448) to R.P., and by Greece and the European Union (European Social Fund-ESF) through the Operational Programme 'Human Resources Development, Education and Lifelong Learning' in the context of the project 'Strengthening Human Resources Research Potential via Doctorate Research' (MIS-5000432), implemented by the Greek State Scholarships Foundation (*IKY*) as a scholarship to G.-R.S.

**Acknowledgements.** We thank Ioannis Livadaras (IMBB/FORTH) for help in the generation of *Drosophila* lines, and Prof. Christos Delidakis (University of Crete and IMBB/FORTH) for providing balancer stocks and useful tips.

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
