## [Reviewer comments · Proceedings of the Royal Society B: Biological Sciences]

Review History

RSPB-2019-2585.R0 (Original submission)

Review form: Reviewer 1 (Shinji Kasai)

Recommendation

Accept with minor revision (please list in comments)

Scientific importance: Is the manuscript an original and important contribution to its field?

Excellent

General interest: Is the paper of sufficient general interest?

Excellent

Quality of the paper: Is the overall quality of the paper suitable?

Good

Is the length of the paper justified?

Yes

Should the paper be seen by a specialist statistical reviewer?

No

Do you have any concerns about statistical analyses in this paper? If so, please specify them explicitly in your report.

No

It is a condition of publication that authors make their supporting data, code and materials available - either as supplementary material or hosted in an external repository. Please rate, if applicable, the supporting data on the following criteria.

Is it accessible?

N/A

Is it clear?

N/A

Is it adequate?

N/A

Do you have any ethical concerns with this paper?

No

Comments to the Author

Transgenic *Drosophila* lines having pyrethroid-metabolizing P450s and sodium channel mutations solely or in combinations were established. Deltamethrin susceptibility and fitness costs of each strain were evaluated. Scope of the study is unique and important to elucidate synergistic effect of multiple number of insecticide-resistance mechanisms. As the authors mention in the paper, as long as I know, this is the first study to isolate the effect of two individual resistance alleles in a common genetic background and examine their combination effect. Recently knockout of resistance genes using CRISPR/Cas9 has become relatively general but editing (knock in) of a resistance mutation is still challenging. In this paper, voltage gated sodium channel, which is the target site of pyrethroid insecticide, is modified by the gene editing technique. Effects of *kdr* genes were evaluated and compared using *Drosophila* lines having the same genetic background. I like the design of the study and as a result, some important facts have been actually elucidated.

Please consider my following comments and suggestions to improve the manuscript.

Major:

1. I understand that the authors used HR-GAL4 driver to express P450 genes in specific tissues related to detoxification. However, we don't know if two different P450 genes were expressed equally in 2 (or 4) lines. I guess the authors should have done any quantitative studies like real time qPCR for mRNA of CYP6BQ23 and CYP9J28. We don't know if CYP6BQ23 is expressed equally in HR-GAL4xUAS-CYP6BQ23 and paraL1014F;HR-GAL4xUAS-CYP6BQ23 (and same for CYP9J28 strains).
2. Line 290 and Table 1: I don't understand the meaning "bearing two copies of each". Does this mean that there are two copies of CYP9J28 as well as HR-GAL4 driver on one chromosome? I don't understand the difference between "homozygous" and "two copies". If my understanding is right, CYP9J28 is expressed two times higher than CYP6BQ23 strains? Please clear this in the Materials and Methods.
3. Results of slow uptake contact bioassay should be expressed as a table like Table 1. I wonder why the authors did not conduct this study for CYP9J28 and V1016G+CYP9J28 strains.
4. Personally, I prefer topical application to bottle assay or filter paper assay because results of the latter two methods sometimes depend on locomotion activity of insects. I wonder if transgenic flies might lose locomotion activity to some extent...
5. Line 336-338: This is an unique and interesting theory. However, in general, P450-generated

metabolites are highly soluble compounds so that I think their affinities to the target site are very weak. Please attach any reference to support the authors theory.

6. Line 346-348: I like this discussion. If pollen beetle possessing both kdr and overexpressed CYP6BQ23 gene is rare in the field due to fitness cost, this is a very important finding.

7. Figure 3: Pupation rates were high in both CYP9J28 and V1016G strains but it was low in V1016G;CYP9J28 strain. I don't understand the reason for why this happened. Please discuss this. Also, I wonder why the authors did not observe oviposition for CYP6BQ23 and CYP9J28 strains which may solve the question; what affected the oviposition activity.

Minor:

Line 116: L1014V□L1016F

Line 242: According to my experience, 1 or 2 day old flies are too young for bioassay. Please show any reference to prove that these age is enough for fly susceptibility to be stable.

Line 245: According to my experience, 1 ul per fly is too much. I wonder this may be 0.1 ul otherwise only 10 flies can be treated using a 10 ul Hamilton syringe. Also, please mention more detail about the system of topical application using 10 ul Hamilton syringe.

Table 1: LC50 should be LD50.

Review form: Reviewer 2

Recommendation

Major revision is needed (please make suggestions in comments)

Scientific importance: Is the manuscript an original and important contribution to its field?

Excellent

General interest: Is the paper of sufficient general interest?

Good

Quality of the paper: Is the overall quality of the paper suitable?

Good

Is the length of the paper justified?

Yes

Should the paper be seen by a specialist statistical reviewer?

No

Do you have any concerns about statistical analyses in this paper? If so, please specify them explicitly in your report.

No

It is a condition of publication that authors make their supporting data, code and materials available - either as supplementary material or hosted in an external repository. Please rate, if applicable, the supporting data on the following criteria.

Is it accessible?

Yes

Is it clear?

Yes

Is it adequate?

Yes

Do you have any ethical concerns with this paper?

No

Comments to the Author

P450-mediated enhanced metabolic detoxification and target-site insensitivity due to mutations in the voltage-gated sodium channel are two major mechanisms of pyrethroid insecticide resistance in various insect pests and human disease vectors. This study took advantage of tools and resources of gene editing in *Drosophila melanogaster*, a model insect, to evaluate potential synergism between the two mechanisms of resistance. They utilized CRISPR-Cas9 and Gal4-UAS to introduce individual and combinations of sodium channel mutations, paraL1014F or paraV1016G, and a pollen beetle P450, Cyp6BQ23 or mosquito Cyp9J28, into a wild-type strain. Their results showed that flies carrying one of the sodium channel mutations or expression of Cyp6BQ23 or Cyp9J28 confers low levels of resistance to pyrethroid deltamethrin, whereas combinations of P450 gene expression and target-site mutations produced a level of resistance that is greater than expected for additive effects. In addition, they found that paraV1016G;HR-GAL4_UAS-CYP9J28 flies (V1016G;CYP9J28), with the dual resistance mechanisms, exhibited a fitness cost in the development time and fecundity, while flies with single resistance loci did not. Overall the approach is sound and the findings are interesting. I have a few specific comments.

- 1) The authors should provide a better rationale for choosing L1014F, V1016G, Cyp6BQ23 or Cyp9J28 and the combinations used for this study.
- 2) It is not clear if they backcrossed their transgenic lines to the parental wild-type line for several generations to generate isogenic lines. If not, the authors should take precautions in interpreting their results on any fitness cost associated with co-existence of these resistance alleles, but not with individual alleles. For example, the figure legend (lines 537-538) indicated that "UAS-CYP9J28 insertion was generated by P-element in random genomic positions, so homozygous flies may bear additional fitness cost unrelated to resistance." Will this issue affect the fitness of paraV1016G;HR-GAL4_UAS-CYP9J28 flies?
- 3) They cited one study supporting their fitness cost theory. However, there are cases where multiple mechanisms of resistance, both enhanced metabolic detoxification and target-site mutations, were co-selected in field populations of agricultural pests and disease vectors with no reported fitness costs. Also there are studies where these mechanisms alone were associated with fitness costs. The authors should give a more comprehensive discussion on this topic considering that this information would be quite important for pest resistance management.
- 4) This approach using a model insect is powerful to confirm the role of resistance-associated mutations and P450 genes from pest species in insecticide resistance and evaluate potential synergistic action between mechanisms. It would be helpful if the authors could discuss pros and cons of using a model insect versus using a pest insect where resistance alleles were initially identified from, such as Brito et al., 2013, PLoS One 8: e60878.

Decision letter (RSPB-2019-2585.R0)

13-Dec-2019

Dear Dr Douris:

I am writing to inform you that your manuscript RSPB-2019-2585 entitled "What I cannot create,

I do not understand": functionally validated synergism of metabolic and target site insecticide resistance" has, in its current form, been rejected for publication in Proceedings B.

This action has been taken on the advice of referees, who have recommended that substantial revisions are necessary. With this in mind we would be happy to consider a resubmission, provided the comments of the referees are fully addressed. However please note that this is not a provisional acceptance.

Sincerely,
Professor Gary Carvalho
mailto:proceedingsb@royalsociety.org

Associate Editor
Board Member: 1
Comments to Author:

In this paper, the authors report the transgenic generation of *Drosophila* lines carrying different resistance alleles to the insecticide deltamethrin from natural populations of *Aedes* mosquitos and *Brassicogethes aeneus* pollen beetles. Phenotypic effects of alleles conferring resistance via two different mechanisms are then investigated both in isolation and in combination, revealing synergistic epistasis in the level of resistance (LC50) conferred for one pair of these alleles. I think quantifying fitness landscapes underlying insecticide resistance in this way, using highly controlled experiments, is an important endeavour, and I agree with the two reviewers that the approach is sound and the results are very interesting. However, I also think there are a number of shortcomings that would have to be have to be addressed before publication, in particular relating to the consistency and statistical analysis of the datasets as well as their presentation and framing in the Background and Discussion sections, as detailed below. The two reviewers also raised a number of issues related to the experimental methods that would have to be addressed.

Specific comments:

- It seems to me that the main achievement of the paper is the construction of the various genotypes that then allows for an exploration of the fitness landscapes underlying resistance and fitness components in the absence of the drug. As also mentioned by reviewer 1, I found it

surprising therefore that some phenotypes (LC50 in slow uptake assays, oviposition) were measured only for a subset of genotypes. I think for consistency and also to take full advantage of the system created, those missing data should be obtained and presented as well.

- I think the statistical analyses need to be improved. Synergistic epistasis in LC50 is one of the key results of the study but not supported by any statistical test. Also, rather surprisingly, 95% fiducial limits are mentioned for LC50 estimates in the methods, but in Table 1 confidence intervals are provided instead. I also think that performing multiple t-tests comparing traits such as LC50 or pupation times between susceptible and resistant genotypes is inappropriate because there are more than just two groups (genotypes). An ANOVA that systematically tests for differences between genotypes would be much more suitable for this kind of data.

- Point 2 raised by reviewer 2 also seems important. As mentioned by the authors in the introduction, the absence of confounding genetic effects is key to the experiments performed here, so it would be good if the authors could confirm that all their lines are isogenic apart from the introduced alleles.

- The Background should be extended with the broad readership of Proceedings B in mind. More information on deltamethrin should be provided (what it is, how it works, where it is used, known resistance mutations & mechanisms, ...), and it would also be good to talk more about epistasis/fitness landscapes in the Introduction - there is a lot about this in the context of antibiotic resistance in bacteria in the literature that should translate to insecticide resistance and could be drawn upon here.

- Similarly, the Discussion needs to be extended; both reviewers make excellent suggestions for important topics to be discussed. If, as a result, the paper becomes too long, some parts of the Methods section could be moved to the Supplementary Material.

- "Co-evolution" is defined as two interacting species evolving in response to each other, and I think the word should not be used in cases where, as in this paper, for cases of polygenic adaptation. E.g., l. 59 "co-evolution of mechanisms" should be replaced by something like "evolution of several resistance mechanisms".

- Figure 2 should be improved - the two plots are inconsistently labelled, font sizes are not ideal, colours are different between the two plots etc. - it simply doesn't look as nice as it could.

Reviewer(s)' Comments to Author:

Referee: 1

Comments to the Author(s)

Transgenic *Drosophila* lines having pyrethroid-metabolizing P450s and sodium channel mutations solely or in combinations were established. Deltamethrin susceptibility and fitness costs of each strain were evaluated. Scope of the study is unique and important to elucidate synergistic effect of multiple number of insecticide-resistance mechanisms. As the authors mention in the paper, as long as I know, this is the first study to isolate the effect of two individual resistance alleles in a common genetic background and examine their combination effect. Recently knockout of resistance genes using CRISPR/Cas9 has become relatively general but editing (knock in) of a resistance mutation is still challenging. In this paper, voltage gated sodium channel, which is the target site of pyrethroid insecticide, is modified by the gene editing technique. Effects of *kdr* genes were evaluated and compared using *Drosophila* lines having the same genetic background. I like the design of the study and as a result, some important facts have been actually elucidated.

Please consider my following comments and suggestions to improve the manuscript.

Major:

1. I understand that the authors used HR-GAL4 driver to express P450 genes in specific tissues related to detoxification. However, we don't know if two different P450 genes were expressed equally in 2 (or 4) lines. I guess the authors should have done any quantitative studies like real time qPCR for mRNA of CYP6BQ23 and CYP9J28. We don't know if CYP6BQ23 is expressed equally in HR-GAL4xUAS-CYP6BQ23 and paraL1014F;HR-GAL4xUAS-CYP6BQ23 (and same for CYP9J28 strains).
2. Line 290 and Table 1: I don't understand the meaning "bearing two copies of each". Does this mean that there are two copies of CYP9J28 as well as HR-GAL4 driver on one chromosome? I don't understand the difference between "homozygous" and "two copies". If my understanding is right, CYP9J28 is expressed two times higher than CYP6BQ23 strains? Please clear this in the Materials and Methods.
3. Results of slow uptake contact bioassay should be expressed as a table like Table 1. I wonder why the authors did not conduct this study for CYP9J28 and V1016G+CYP9J28 strains.
4. Personally, I prefer topical application to bottle assay or filter paper assay because results of the latter two methods sometimes depend on locomotion activity of insects. I wonder if transgenic flies might lose locomotion activity to some extent...
5. Line 336-338: This is an unique and interesting theory. However, in general, P450-generated metabolites are highly soluble compounds so that I think their affinities to the target site are very weak. Please attach any reference to support the authors theory.
6. Line 346-348: I like this discussion. If pollen beetle possessing both kdr and overexpressed CYP6BQ23 gene is rare in the field due to fitness cost, this is a very important finding.
7. Figure 3: Pupation rates were high in both CYP9J28 and V1016G strains but it was low in V1016G;CYP9J28 strain. I don't understand the reason for why this happened. Please discuss this. Also, I wonder why the authors did not observe oviposition for CYP6BQ23 and CYP9J28 strains which may solve the question; what affected the oviposition activity.

Minor:

Line 116: L1014V□L1016F

Line 242: According to my experience, 1 or 2 day old flies are too young for bioassay. Please show any reference to prove that these age is enough for fly susceptibility to be stable.

Line 245: According to my experience, 1 ul per fly is too much. I wonder this may be 0.1 ul otherwise only 10 flies can be treated using a 10 ul Hamilton syringe. Also, please mention more detail about the system of topical application using 10 ul Hamilton syringe.

Table 1: LC50 should be LD50.

Referee: 2

Comments to the Author(s)

P450-mediated enhanced metabolic detoxification and target-site insensitivity due to mutations in the voltage-gated sodium channel are two major mechanisms of pyrethroid insecticide resistance in various insect pests and human disease vectors. This study took advantage of tools and resources of gene editing in *Drosophila melanogaster*, a model insect, to evaluate potential synergism between the two mechanisms of resistance. They utilized CRISPR-Cas9 and Gal4-UAS to introduce individual and combinations of sodium channel mutations, paraL1014F or paraV1016G, and a pollen beetle P450, Cyp6BQ23 or mosquito Cyp9J28, into a wild-type strain. Their results showed that flies carrying one of the sodium channel mutations or expression of Cyp6BQ23 or Cyp9J28 confers low levels of resistance to pyrethroid deltamethrin, whereas combinations of P450 gene expression and target-site mutations produced a level of resistance that is greater than expected for additive effects. In addition, they found that paraV1016G;HR-GAL4_UAS-CYP9J28 flies (V1016G;CYP9J28), with the dual resistance mechanisms, exhibited a fitness cost in the development time and fecundity, while flies with single resistance loci did not. Overall the approach is sound and the findings are interesting. I have a few specific comments.

- 1) The authors should provide a better rationale for choosing L1014F, V1016G, Cyp6BQ23 or Cyp9J28 and the combinations used for this study.
- 2) It is not clear if they backcrossed their transgenic lines to the parental wild-type line for several generations to generate isogenic lines. If not, the authors should take precautions in interpreting their results on any fitness cost associated with co-existence of these resistance alleles, but not with individual alleles. For example, the figure legend (lines 537-538) indicated that “UAS-CYP9J28 insertion was generated by P-element in random genomic positions, so homozygous flies may bear additional fitness cost unrelated to resistance.” Will this issue affect the fitness of paraV1016G;HR-GAL4_UAS-CYP9J28 flies?
- 3) They cited one study supporting their fitness cost theory. However, there are cases where multiple mechanisms of resistance, both enhanced metabolic detoxification and target-site mutations, were co-selected in field populations of agricultural pests and disease vectors with no reported fitness costs. Also there are studies where these mechanisms alone were associated with fitness costs. The authors should give a more comprehensive discussion on this topic considering that this information would be quite important for pest resistance management.
- 4) This approach using a model insect is powerful to confirm the role of resistance-associated mutations and P450 genes from pest species in insecticide resistance and evaluate potential synergistic action between mechanisms. It would be helpful if the authors could discuss pros and cons of using a model insect versus using a pest insect where resistance alleles were initially identified from, such as Brito et al., 2013, PLoS One 8: e60878.

Author's Response to Decision Letter for (RSPB-2019-2585.R0)

See Appendix A.

RSPB-2020-0838.R0

Review form: Reviewer 2

Recommendation

Accept as is

Scientific importance: Is the manuscript an original and important contribution to its field?

Good

General interest: Is the paper of sufficient general interest?

Good

Quality of the paper: Is the overall quality of the paper suitable?

Good

Is the length of the paper justified?

Yes

Should the paper be seen by a specialist statistical reviewer?

Yes

Do you have any concerns about statistical analyses in this paper? If so, please specify them explicitly in your report.

No

It is a condition of publication that authors make their supporting data, code and materials available - either as supplementary material or hosted in an external repository. Please rate, if applicable, the supporting data on the following criteria.

Is it accessible?

Yes

Is it clear?

Yes

Is it adequate?

Yes

Do you have any ethical concerns with this paper?

No

Comments to the Author

The authors have addressed the concerns raised by the reviewers. The revised manuscript is significantly improved.

Decision letter (RSPB-2020-0838.R0)

01-May-2020

Dear Dr Douris

I am pleased to inform you that your manuscript RSPB-2020-0838 entitled "What I cannot create, I do not understand": functionally validated synergism of metabolic and target site insecticide resistance" has been accepted for publication in Proceedings B.

The referee(s) have recommended publication, but also suggest some minor revisions to your manuscript. Therefore, I invite you to respond to the referee(s)' comments and revise your manuscript. Because the schedule for publication is very tight, it is a condition of publication that you submit the revised version of your manuscript within 7 days. If you do not think you will be able to meet this date please let us know.

When submitting your revised manuscript, you will be able to respond to the comments made by the referee(s) and upload a file "Response to Referees". You can use this to document any changes you make to the original manuscript. We require a copy of the manuscript with revisions made

since the previous version marked as 'tracked changes' to be included in the 'response to referees' document.

Sincerely,
Professor Gary Carvalho
mailto: proceedingsb@royalsociety.org

Associate Editor
Board Member
Comments to Author:

This resubmission has been assessed by the more critical of the original two reviewers. This reviewer was happy with the improvements that the authors made in response to the issues raised, and I agree that the authors have done a good job revising their ms. I have spotted a few minor problems (listed below) but I think the authors can be trusted to fix those in their final version.

Abstract: It would be good to mention in the first sentence that we're talking about insects here, e.g.: "The putative synergistic action of target-site mutations and enhanced detoxification in pyrethroid resistance in insects has been ..."

l.81: "have been long been"  "have long been"
l.411: closing bracket and full stop missing

Reviewer(s)' Comments to Author:

Referee: 2

Comments to the Author(s).
The authors have addressed the concerns raised by the reviewers. The revised manuscript is significantly improved.

Author's Response to Decision Letter for (RSPB-2020-0838.R0)

See Appendix B.

Decision letter (RSPB-2020-0838.R1)

04-May-2020

Dear Dr Douris

I am pleased to inform you that your manuscript entitled "What I cannot create, I do not understand": functionally validated synergism of metabolic and target site insecticide resistance" has been accepted for publication in Proceedings B.

Open Access

Paper charges

Sincerely,

Appendix A

Please find below our response to all comments raised by the Associate editor and the referees regarding the submission of our manuscript “*What I cannot create, I do not understand*”: *functionally validated synergism of metabolic and target site insecticide resistance*, with Manuscript ID RSPB-2019-2585.

Author responses are shown in red.

Associate Editor

Board Member: 1

Comments to Author:

In this paper, the authors report the transgenic generation of Drosophila lines carrying different resistance alleles to the insecticide deltamethrin from natural populations of Aedes mosquitos and Brassicogethes aeneus pollen beetles. Phenotypic effects of alleles conferring resistance via two different mechanisms are then investigated both in isolation and in combination, revealing synergistic epistasis in the level of resistance (LC50) conferred for one pair of these alleles. I think quantifying fitness landscapes underlying insecticide resistance in this way, using highly controlled experiments, is an important endeavour, and I agree with the two reviewers that the approach is sound and the results are very interesting. However, I also think there are a number of shortcomings that would have to be addressed before publication, in particular relating to the consistency and statistical analysis of the datasets as well as their presentation and framing in the Background and Discussion sections, as detailed below. The two reviewers also raised a number of issues related to the experimental methods that would have to be addressed.

We thank the Associate editor and reviewers for their assessment. We have made considerable effort, given the current circumstances, to address all issues raised, which we believe has considerably improved the research presented.

Specific comments:

- It seems to me that the main achievement of the paper is the construction of the various genotypes that then allows for an exploration of the fitness landscapes underlying resistance and fitness components in the absence of the drug. As also mentioned by reviewer 1, I found it surprising therefore that some phenotypes (LC50 in slow uptake assays, oviposition) were measured only for a subset of genotypes. I think for consistency and also to take full advantage of the system created, those missing data should be obtained and presented as well.

The editor is correct to point out that some of the phenotypes were only measured for a subset of genotypes. Therefore, we have updated the manuscript to include both the LC₅₀ slow uptake toxicology data for the *A. aegypti* genotypes (previously only the *M. anaeus* genotypes were included). These data can now be found in Table S3. Additionally, we have included all of the genotypes for the oviposition data (Figure 3). In summary, the data has now become symmetrical; all genotypes were used in all assays.

- I think the statistical analyses need to be improved. Synergistic epistasis in LC50 is one of the key results of the study but not supported by any statistical test. Also, rather surprisingly, 95% fiducial limits are mentioned for LC50 estimates in the methods, but in Table 1 confidence intervals are provided instead.

Indeed, LC50s have fiducial limits not confidence intervals, and the numbers in Table 1 correspond to fiducial limits. This was a wording error; we have updated the legend accordingly.

Regarding statistical support of LC₅₀ values, since these are already a statistical transformation the standard practice in insect toxicology literature is to perform a chi-square test for linearity (which is done by default by PoloPlus software, and we do) and to consider significant the LC₅₀ values and RR (resistance ratio) if the 95% fiducial limits do not include 1 (following for example Robertson, 2008). We provide the relevant information in the Methods section and relevant references.

I also think that performing multiple t-tests comparing traits such as LC50 or pupation times between susceptible and resistant genotypes is inappropriate because there are more than just two groups (genotypes). An ANOVA that systematically tests for differences between genotypes would be much more suitable for this kind of data.

This is a very good point and we thank the editor for bringing it up. We have performed a one-way ANOVA analysis for each dataset and updated Figure 3 to indicate the results. The statistical significance (p-values) from each analysis is provided in Tables S4 and S5. The original fitness data for all life-table parameters measured are included in Supplementary Dataset 1 as an excel file.

- Point 2 raised by reviewer 2 also seems important. As mentioned by the authors in the introduction, the absence of confounding genetic effects is key to the experiments performed here, so it would be good if the authors could confirm that all their lines are isogenic apart from the introduced alleles.

The authors agree that this is a valid point. The lines are not completely isogenic, so the authors have updated the manuscript to indicate this at several locations. We provide precise information on the genetic makeup of each strain in the Methods text and collectively in Table S1. We have also updated the bioassay results to include controls from both strains or crosses (nos.Cas9 and HR-GAL4 x yw;attP40) that contribute to the genetic background of the strains bearing resistance alleles, and demonstrate that there is no difference at their resistance levels at topical bioassays (Table 1) that could possibly account for the differences observed by synergistic interactions among resistance alleles. Thus, we maintain that this study represents a major improvement over prior attempts to examine synergism in terms of genetic homogeneity, but to be more accurate we now frame it as a major improvement rather than an ideal genetic control.

- The Background should be extended with the broad readership of Proceedings B in mind. More information on deltamethrin should be provided (what it is, how it works, where it is used, known resistance mutations & mechanisms, ...), and it would also be good to talk more about epistasis/fitness landscapes in the Introduction - there is a lot about this in the context of antibiotic resistance in bacteria in the literature that should translate to insecticide resistance and could be drawn upon here.

The authors have addressed these concerns by adding in an additional paragraph to contextualize pyrethroids and the resistance mechanisms associated with them. This should help to address the logic behind choosing these compounds for a more general audience. We have also added a few extra paragraphs on fitness cost and epistasis.

- Similarly, the Discussion needs to be extended; both reviewers make excellent suggestions for important topics to be discussed. If, as a result, the paper becomes too long, some parts of the Methods section could be moved to the Supplementary Material.

To address these concerns the authors have extended the discussion significantly. In particular, we have added sections on *Drosophila* as a model organisms and increased our discussion on metabolite toxicity and fitness costs.

- "Co-evolution" is defined as two interacting species evolving in response to each other, and I think the word should not be used in cases where, as in this paper, for cases of polygenic adaptation. E.g., l. 59 "co-evolution of mechanisms" should be replaced by something like "evolution of several resistance mechanisms".

The authors have removed the term co-evolution from the manuscript in order to avoid any confusion.

- Figure 2 should be improved - the two plots are inconsistently labelled, font sizes are not ideal, colours are different between the two plots etc. - it simply doesn't look as nice as it could.

The authors have updated Figure 2 by adding qPCR results requested (see below), and have also completely replaced Figure 3 with a new figure that we believe give a much better representation of the fitness data and is much more in line with the journal standards.

Reviewer(s)' Comments to Author:

Referee: 1

Comments to the Author(s)

Transgenic Drosophila lines having pyrethroid-metabolizing P450s and sodium channel mutations solely or in combinations were established. Deltamethrin susceptibility and fitness costs of each strain were evaluated. Scope of the study is unique and important to elucidate synergistic effect of multiple number of insecticide-resistance mechanisms. As the authors mention in the paper, as long as I know, this is the first study to isolate the effect of two individual resistance alleles in a common genetic background and examine their combination effect. Recently knockout of resistance genes using CRISPR/Cas9 has become relatively general but editing (knock in) of a resistance mutation is still challenging. In this paper, voltage gated sodium channel, which is the target site of pyrethroid insecticide, is modified by the gene editing technique. Effects of kdr genes were evaluated and compared using Drosophila lines having the same genetic background. I like the design of the study and as a result, some important facts have been actually elucidated.

Please consider my following comments and suggestions to improve the manuscript.

Major:

1. I understand that the authors used HR-GAL4 driver to express P450 genes in specific tissues related to detoxification. However, we don't know if two different P450 genes were expressed equally in 2 (or 4) lines. I guess the authors should have done any quantitative studies like real time qPCR for mRNA of CYP6BQ23 and CYP9J28. We don't know if CYP6BQ23 is expressed equally in HR-GAL4xUAS-CYP6BQ23 and paraL1014F;HR-GAL4xUAS-CYP6BQ23 (and same for CYP9J28 strains).

We appreciate this point. We have performed qPCR to quantify expression of different strains and crosses and have not found any significant difference in expression levels among strains expressing P450 transgenes in the presence or absence of the relevant mutation. The qPCR results for both P450s in either wild-type or genome modified background are now included in Figure 2 (panel C). A relevant section has also been added at the Methods.

2. Line 290 and Table 1: I don't understand the meaning "bearing two copies of each". Does this mean that there are two copies of CYP9J28 as well as HR-GAL4 driver on one chromosome? I don't understand the difference between "homozygous" and "two copies". If my understanding is right, CYP9J28 is expressed two times higher than CYP6BQ23 strains? Please clear this in the Materials and Methods.

We thank the reviewer for letting us clarify this point. As we now make clearer in the Methods section, we generated a strain (yw;HR-GAL4>UAS-CYP9J28) which bears two copies of each transgene when the recombinant chromosome bearing both transgenes is homozygous, in order to obtain readily detectable levels of resistance in topical application bioassays. We have modified the text to make this point clearer even to the non-specialist.

The expression levels between CYP9J28 and CYP6BQ23 are not readily comparable however, since these transgenes have different insertion points in the genome and their relative expression level is also influenced by different position effects.

3. Results of slow uptake contact bioassay should be expressed as a table like Table 1. I wonder why the authors did not conduct this study for CYP9J28 and V1016G+CYP9J28 strains.

As stated previously, a full array of bioassay results is now included in the manuscript. Slow uptake contact bioassays are collectively shown in Table S3 and presented also in the Results section.

4. Personally, I prefer topical application to bottle assay or filter paper assay because results of the latter two methods sometimes depend on locomotion activity of insects. I wonder if transgenic flies might lose locomotion activity to some extent...

The authors have performed both assays to avoid any shortcomings of only considering one. In general, we did not observe any locomotion activity loss in the most resistant flies even at the highest doses with the contact (slow uptake) bioassay.

5. Line 336-338: This is an unique and interesting theory. However, in general, P450-generated metabolites are highly soluble compounds so that I think their affinities to the target site are very weak. Please attach any reference to support the authors theory.

The authors have addressed these concerns by providing additional evidence (please see Discussion, lines 416-427):

"Alternatively, the less toxic P450-generated metabolites might bind the mutant target receptors disproportionately less effectively, thus manifesting in a synergistic phenotype. It is currently unknown what, if any, affinity the P450 derived metabolites of deltamethrin would have for para, but it has been suggested that their accumulation is likely to be detrimental in mosquitoes and that their metabolism by certain P450s, such as the Anopheles gambiae CYPZ8 and CYP6Z2, is a secondary but important mechanism of insecticide resistance (Chandor-Proust et al 2013). In addition, examples from other compounds may provide some guidance. Imidacloprid is "detoxified" by P450s into a variety of

metabolites which still show levels of toxicity that would probably be relevant in vivo [36,37]. Nevertheless, there is currently no data regarding the binding of various pyrethroid metabolites for different alleles of para, so such a hypothesis awaits functional validation”)

6. Line 346-348: I like this discussion. If pollen beetle possessing both *kdr* and overexpressed *CYP6BQ23* gene is rare in the field due to fitness cost, this is a very important finding.

The authors appreciate the reviewer’s interest.

7. Figure 3: Pupation rates were high in both *CYP9J28* and *V1016G* strains but it was low in *V1016G;CYP9J28* strain. I don’t understand the reason for why this happened. Please discuss this. Also, I wonder why the authors did not observe oviposition for *CYP6BQ23* and *CYP9J28* strains which may solve the question; what affected the oviposition activity.

We have now updated the fitness analysis to include all strains tested (Figure 3 and Tables S4 and S5) and have updated Figure 3, though the original data were also present in the Supplementary Dataset 1. However, in contrast to synergism in terms of resistance, the authors do not have any explanation for any kind of synergistic mechanism that translates to fitness cost. However, we have updated the discussion of the fitness cost to highlight the lack of a firm mechanism to explain fitness interactions and have also provided more details in general. This point is indeed important and open to future research.

Minor:

Line 116: *L1014V*à*L1016F*

Done

Line 242: According to my experience, 1 or 2 day old flies are too young for bioassay. Please show any reference to prove that these age is enough for fly susceptibility to be stable.

We provide certain references where 1-3 day old flies are used for bioassays (Daborn et al., 2007, Samantsidis et al., 2019). We routinely use this strategy in bioassays against several insecticide classes and have not observed any significant differentiation between day 1 or day 3-4 flies.

Line 245: According to my experience, 1 ul per fly is too much. I wonder this may be 0.1 ul otherwise only 10 flies can be treated using a 10 ul Hamilton syringe. Also, please mention more detail about the system of topical application using 10 ul Hamilton syringe.

Although we routinely use 1 µl for several insect species (mosquitoes, olive fruit flies and *Drosophila*), admittedly *Drosophila* is somewhat smaller and the volume of the application may be a concern. However, we have not observed any particular stress of the flies in our experiments, and this is supported by the fact that there is no additional mortality observed in the control experiments. The flies are indeed treated in batches of 10.

Table 1: LC50 should be LD50.

This was corrected; we thank the reviewer for pointing it out.

Referee: 2

Comments to the Author(s)

P450-mediated enhanced metabolic detoxification and target-site insensitivity due to mutations in the voltage-gated sodium channel are two major mechanisms of pyrethroid insecticide resistance in various insect pests and human disease vectors. This study took advantage of tools and resources of gene editing in *Drosophila melanogaster*, a model insect, to evaluate potential synergism between the two mechanisms of resistance. They utilized CRISPR-Cas9 and Gal4-UAS to introduce individual and combinations of sodium channel mutations, *para*L1014F or *para*V1016G, and a pollen beetle P450, *Cyp6BQ23* or mosquito *Cyp9J28*, into a wild-type strain. Their results showed that flies carrying one of the sodium channel mutations or expression of *Cyp6BQ23* or *Cyp9J28* confers low levels of resistance to pyrethroid deltamethrin, whereas combinations of P450 gene expression and target-site mutations produced a level of resistance that is greater than expected for additive effects. In addition, they found that *para*V1016G;*HR-GAL4_UAS-CYP9J28* flies (V1016G;*CYP9J28*), with the dual resistance mechanisms, exhibited a fitness cost in the development time and fecundity, while flies with single resistance loci did not.

Overall the approach is sound and the findings are interesting. I have a few specific comments.

1) *The authors should provide a better rationale for choosing L1014F, V1016G, Cyp6BQ23 or Cyp9J28 and the combinations used for this study.*

The authors have updated the introduction in order to include a more thorough justification for why deltamethrin was chosen and the reason that these particular genetic factors were selected.

2) *It is not clear if they backcrossed their transgenic lines to the parental wild-type line for several generations to generate isogenic lines. If not, the authors should take precautions in interpreting their results on any fitness cost associated with co-existence of these resistance alleles, but not with individual alleles. For example, the figure legend (lines 537-538) indicated that “UAS-CYP9J28 insertion was generated by P-element in random genomic positions, so homozygous flies may bear additional fitness cost unrelated to resistance.” Will this issue affect the fitness of *para*V1016G;*HR-GAL4_UAS-CYP9J28* flies?*

The authors understand how this may have been a point of confusion. The lines have not been backcrossed and are thus not fully isogenic. However, during our crossing strategy we attempted to reduce the amount of background genetic variation between the control lines and the various lines carrying resistance factors. We now provide extensive information on the genetic makeup of each strain and argue that while the genetic background of the strains contributing to the generation of the strains used for this study are not fully equivalent, their resistance properties are not significantly different.

Regarding the fact that UAS-CYP9J28 has been generated by P-element insertion and is thus less controllable than UAS-CYP6BQ23 that is in attP40 position, one would expect the potential cost of UAS-CYP9J28 insertion to manifest in all genotypes that contain this insertion (at least the homozygotes). However, ANOVA of fitness data (Figure 3 and Tables S4 and S5) indicates that *para*^{V1016G};*HR-GAL4>UAS-CYP9J28*(2N) flies differ significantly both in developmental time and fecundity not only from background control strains (nos.Cas9 and *HR-GAL4 x yw;attP40*) or genome modifies strains not bearing UAS-CYP9J28 insertion (*para*^{V1016G}) but also from the strain bearing the randomly inserted transgene (*yw; HR-GAL4>UAS-CYP9J28*). Thus, we are confident that the fitness cost observed in *para*^{V1016G};*HR-GAL4>UAS-CYP9J28*(2N) flies is associated with the synergistic

interaction rather than the genetic makeup of the specific strain, although ideally one would prefer to have a perfectly controlled genetic background to fully investigate the impact on fitness.

We have thus edited the manuscript at several points in the introduction and discussion to reflect these considerations. Furthermore, we have changed the text to be more reserved about the interpretation of our results. We maintain that this study represents a major improvement over prior attempts to examine synergism in terms of genetic homogeneity, but we more cautiously now frame it as an improvement rather than an ideal genetic control.

3) *They cited one study supporting their fitness cost theory. However, there are cases where multiple mechanisms of resistance, both enhanced metabolic detoxification and target-site mutations, were co-selected in field populations of agricultural pests and disease vectors with no reported fitness costs. Also there are studies where these mechanisms alone were associated with fitness costs. The authors should give a more comprehensive discussion on this topic considering that this information would be quite important for pest resistance management.*

The authors have significantly increased the discussion of fitness cost in both the discussion and the introduction. Furthermore, we have included additional data in the results providing more evidence to support the fitness cost section of the discussion.

4) *This approach using a model insect is powerful to confirm the role of resistance-associated mutations and P450 genes from pest species in insecticide resistance and evaluate potential synergistic action between mechanisms. It would be helpful if the authors could discuss pros and cons of using a model insect versus using a pest insect where resistance alleles were initially identified from, such as Brito et al., 2013, PLoS One 8: e60878.*

The authors have modified the discussion to include comparisons of model organisms versus pest insects and discuss the suggested very useful reference. Briefly, the authors suggest that *Drosophila* allows for the combination of different genetic factors in a way that requires less effort than pest species. However, we avoid a very extensive discussion of pros and cons of *Drosophila* vs pest species since this is the subject of relevant recent review articles (including from our own team, to appear in a forthcoming issue of Pesticide Biochemistry and Physiology) where the reader is referred to for further details.

Appendix B

Please find below our response to all comments raised regarding the submission of our manuscript *“What I cannot create, I do not understand”*: *functionally validated synergism of metabolic and target site insecticide resistance*, with Manuscript ID RSPB-2020-0838.

Author responses are shown in red.

This resubmission has been assessed by the more critical of the original two reviewers. This reviewer was happy with the improvements that the authors made in response to the issues raised, and I agree that the authors have done a good job revising their ms.

We thank the reviewer for his kind assessment

I have spotted a few minor problems (listed below) but I think the authors can be trusted to fix those in their final version.

Abstract: It would be good to mention in the first sentence that we're talking about insects here, e.g.: "The putative synergistic action of target-site mutations and enhanced detoxification in pyrethroid resistance in insects has been ..."

Done

l.81: "have been long been"  "have long been"

Done

l.411: closing bracket and full stop missing

Done

Additional changes in the final version of the manuscript:

1. The doi was added in reference #33
2. Figure 2C was updated to better indicate standard deviation at the left panel

A “track-changes” version of the final text that includes the changes made follows below: